# Immobilization and Release of Platelet-Rich Plasma from Modified Nanofibers Studied by Advanced X-ray Photoelectron Spectroscopy Analyses

**DOI:** 10.3390/polym15061440

**Published:** 2023-03-14

**Authors:** Anton M. Manakhov, Elizaveta S. Permyakova, Anastasiya O. Solovieva, Natalya A. Sitnikova, Philipp V. Kiryukhantsev-Korneev, Anton S. Konopatsky, Dmitry V. Shtansky

**Affiliations:** 1Research Institute of Clinical and Experimental Lymphology—Branch of the ICG SB RAS, 2 Timakova St., 630060 Novosibirsk, Russia; 2National University of Science and Technology “MISiS”, Leninsky Prospekt 4, 119049 Moscow, Russia

**Keywords:** platelet-rich plasma, XPS, immobilization, nanofibers, surface chemistry

## Abstract

Platelet-rich Plasma (PRP) is an ensemble of growth factors, extracellular matrix components, and proteoglycans that are naturally balanced in the human body. In this study, the immobilization and release of PRP component nanofiber surfaces modified by plasma treatment in a gas discharge have been investigated for the first time. The plasma-treated polycaprolactone (PCL) nanofibers were utilized as substrates for the immobilization of PRP, and the amount of PRP immobilized was assessed by fitting a specific X-ray Photoelectron Spectroscopy (XPS) curve to the elemental composition changes. The release of PRP was then revealed by measuring the XPS after soaking nanofibers containing immobilized PRP in buffers of varying pHs (4.8; 7.4; 8.1). Our investigations have proven that the immobilized PRP would continue to cover approximately fifty percent of the surface after eight days.

## 1. Introduction

The sequential process of wound healing, which includes inflammation, re-epithelialization, angiogenesis, granular tissue creation, wound closure, and the production of normal tissue, is well understood. It involves the interaction of many cell types, soluble mediators, and extracellular matrices [1] The violation of this cascade, including secondary infections, leads to ineffective healing, chronicity of the process, and the rejection of skin grafts [2].

The extracellular matrix (ECM) regulates cell migration, proliferation, and angiogenesis by acting as a signaling chemical reservoir, providing structural support, intercellular interaction, and angiogenesis. Fibroblasts’ secreted polysaccharides, proteoglycans, and fibrin fibers form a three-dimensional structure [3,4]. The creation of bioengineered skin substitutes aims to mimic the native ECM’s qualities, which are crucial to the course of normal healing. The development of matrices for the treatment of skin injuries must take into account factors including biocompatibility, extracellular matrix-like structure, atraumatic nature, stimulation of angiogenesis, and antibacterial activities.

Effective neovasculogenesis is a crucial component of wound healing, hence improving the angiogenic capacity of biomaterials is a goal for their development. Vascular endothelial growth factors [5], a mixture of factors [6,7], chitosan derivatives with heparin, amino acids [8], copper ions, and other components are included in the biocomposite composition for this purpose [9,10,11].

Tissue engineering structures are constructed using biomaterials, which are organic or inorganic compounds that resemble the extracellular matrix. Collagen, chitosan, alginate polypeptides, hyaluronans, glycosaminoglycans, fibronectin, etc. are a few examples of natural polymers [12,13]. Natural polymers can cause immunogenic responses, have variable rates of degradation, and have limited capacity for modification. Polyglycolide, polylactide, polytetrafluoroethylene, polyethylene terephthalate, and polycaprolactone are examples of synthetic biodegradable materials.

Tissue engineering structures based on bionanomaterials are now being actively developed. Synthetic polymers can be easily manipulated to create a regulated concentration and release rate of growth factors and/or antimicrobial chemicals. They are also less expensive, more homogeneous, and do not have immunogenic effects. Additionally, the fabrication of nanoparticles and the development of a technology that enables the modification and customization of the material’s structure and physicochemical properties under the circumstances of wound healing make nanomaterials ubiquitous. It should be highlighted that untreated synthetic polymer nanofibers, such as polycaprolactone and polylactide, have poor cell adherence and slow wound healing [14,15]. Additionally, it seems almost impossible for active biomolecules (such as antibiotics and growth factors) to adhere to untreated nanofibers.

The electrospinning of biodegradable nanofibers from a solution of polymers (polycaprolactone, polyethylene glycol, polylactide, etc.) is the most promising technique [16]. The Nanospider method created by Elmarco (Elmarco, Libec, Czech Republic) at the same time made it possible to produce enormous nanofiber films (30 cm wide and any length) [17,18]. This method produces nanofibers with a regulated morphology and a low cost (the structure is depicted in the attached extra information file). Collagen, gelatin, and chitosan are examples of natural polymers that can be used in a solution to electrospin nanofibers (polycaprolactone, polyethylene glycol, etc.).

Although natural polymer-derived nanofibers exhibit great biocompatibility, it can be challenging to produce stable, homogenous nanofibers of these materials [19]. It was impossible to obtain pure collagen and chitosan nanofibers for a long period; as a result, fibers made up of a blend of polymers, such as chitosan/polyethylene oxide, were produced [16]. Collagen is also a very expensive product, and gelatin and collagen nanofibers frequently degrade in aqueous environments. Consequently, creating bioactive nanofibers from synthetic polymers is a very promising endeavor. However, as the majority of these polymeric nanofibers have a high water contact angle (are superhydrophobic), additional processing is required to improve the cell adhesion and proliferation on such surfaces [20].

Currently, co-spinning of biopolymers (such as gelatin or collagen) with nanofibers, plasma treatment in a gas discharge combined with (or without) grafting growth factors, and plasma polymerization are the four most popular ways to modify nanofibers [21]. Since liquid processing causes nanofibers to degrade, the first option is the least promising. Because the surface of the nanofibers lacks active groups that may be further connected with active chemicals (such as growth factors or antibiotics), the second technique has the drawbacks of being non-universal and having too few application options. The third way is the most energy-effective: plasma treatment (treatment in a gas discharge such as air, oxygen, or argon) [22,23].

Wound healing has been markedly enhanced by the attachment of growth factors to plasma-treated polycaprolactone and polylactide nanofibers [24]. While it is currently uncertain how long the action of immobilized growth factors may be maintained on the surface, the effect of plasma treatment is lost very rapidly (within 1–2 days); hence growth factors should be applied immediately after plasma treatment. The latter technique (plasma polymerization, or the deposition of plasma polymers caused by a discharge in organic monomer vapor), which is also energy-efficient, is the most promising since the plasma polymer layers that are formed are extremely stable [19]. In addition, plasma polymers always have a larger concentration of active surface groups than plasma-treated surfaces. However, this method of processing nanofibers is the least studied. Additionally, the plasma-coated PCL nanofibers with immobilized PRP can be used to treat diabetic wounds [25,26].

By adding COOH groups to the surface of nanofibers, it is feasible to bind protein molecules through the creation of a covalent bond, in addition to improving the biocompatibility (decreasing hydrophobicity) [27].

Mesenchymal stem cells adhered and proliferated more strongly to the plasma-modified PCL nanofibers containing COOH groups. Still, PCL nanofibers with covalently bound bioactive compounds from platelet-rich plasma (PRP) had the greatest efficacy [25,27].

PRP is an ensemble of growth factors, extracellular matrix components, and proteoglycans that is naturally balanced. It is composed of platelet-derived growth factors (PDGF-AA, BB, and AB-isomers), transforming factor growth- (TGF-), platelet factor 4 (PF4), interleukin-1 (IL-1), platelet-derived angiogenesis factor (PDAF), vascular endothelial growth factor (VEGF), epidermal growth factor (EGF), epithelial growth factor cells (ECGFs), insulin-like growth factor (IGF), osteocalcin (TSP-1).

The “reservoir” of PRP immobilized on a COOH plasma polymer layer may enhance the proliferation and migration of stem cells, attract macrophages, regulate the wound’s cytokine backdrop, and restrict inflammation. Promoting the formation of new capillaries accelerates epithelization in chronic wounds of various etiologies, hence enhancing wound healing.

The immobilization of biomolecules onto plasma-modified surfaces is quite challenging to control. Despite the fact that the immobilization can be qualitatively confirmed by some changes in the spectra (e.g., by IR or X-ray Photoelectron Spectrospies, XPS), the quantitative analysis is challenging. In our previous works, we performed an attempt to model the XPS spectra and to calculate the amount of the proteins with well-known structures (Apoliprotein A1 and Angeigenin) [28]. However, if one is planning to model more complex bioactive substances, such as PRP, that approach will not be valid.

In this work, for the first time, an investigation of the immobilization and release of PRP for plasma-modified nanofibers has been performed. The plasma-coated PCL nanofibers were used as substrates for the immobilization of PRP, and the amount of PRP immobilized was quantified by using a special XPS curve fitting coupled with the changes in the elemental composition. Then, the release of PRP was controlled by measuring XPS after soaking the nanofibers with immobilized PRP in buffers with different pHs. Our research has shown that even after 8 days, the immobilized PRP would still cover more than 50% of the surface.

## 2. Materials and Methods

### 2.1. Preparation of Nanofibers

Nanofibers were produced by electrospinning a 9-weight percent solution of PCL (80,000 g/mol) solution. The sample processing can be found elsewhere [29]. Briefly, acetic acid (99%) and formic acid (98%) were used to dissolve the granulated PCL. All substances were bought from Sigma Aldrich (Darmstadt, Germany). The acetic acid (AA) and formic acid (FA) had a weight ratio of 2:1. The samples were electrospun by a Super ES-2 machine produced by ESpin Nanotech, (ESpin Nanotech, Kanpur, India), which included both drum and static plate collectors. In this investigation, a static collector plate has been used for collecting the nanofibers. The flowrate of PCL solution was 1 mL/h. The samples were collected onto polypropylene fabric and placed at 12 cm distance from the nozzle. The electrospinining voltage was kept at 50 kV. The authors used the term PCL-ref to refer to the untreated, as-prepared PCL nanofibers.

### 2.2. Plasma-Coated COOH

The plasma polymerization technique for Ar/CO_2_/C_2_H_4_ is discussed in full elsewhere [30,31]. On Si wafers and PCL nanofibers, COOH plasma polymer layers were deposited utilizing a UVN-2M vacuum system equipped with rotary and oil diffusion pumps. The reactor’s residual pressure was below 10^−3^ Pa. A radio frequency (RF) Cito 1310-ACNA-N37A-FF power supply (Comet, Flamatt, Switzerland) connected to an RFPG-128 disk generator (Beams and Plasmas, Moscow, Russia) located in the vacuum chamber was used to ignite the plasma. The duty cycle and RF power were, respectively, set to 5% and 500 W. In the vacuum chamber, CO_2_ (99.995%), Ar (99.998%), and C_2_H_4_ (99.95%) were introduced. The Ar gas flow was fixed at 50 sccm, whereas the CO_2_ and C_2_H_4_ gas fluxes were set to 35 and 10 sccm, respectively. They were managed with the aid of a 647C Multi-Gas Controller (MKST, Newport, RI, USA). Using a VMB-14 unit (Tokamak Company, Dubna, Russia) and D395-90-000 BOC (Edwards Vacuum, Sanborn, NY, USA) controllers, the chamber pressure was measured. In total, 8 cm was established as the distance between the RF electrode and the substrate. The time allotted for the deposition was 15 min. The plasma-coated nanofibers were denoted as PCL-COOH.

### 2.3. Sample Characterization

The morphology of the material was studied using scanning electron microscopy (SEM). A JEOL Ltd. JSMF 7600 (JEOL, Tokyo, Japan) microscope equipped with an energy-dispersive X-ray spectrometer was utilized for SEM investigation. To compensate for surface charge, a 5 nm-thick Pt coating was applied to the samples.

X-ray photoelectron spectroscopy (XPS), was used to characterize the chemical composition of the sample. The XPS examination was conducted with a PHI5500VersaProbeII (Ulvac PHI, Osaka, Japan) instrument equipped with a monochromatic Al K X-ray source (h = 1486.6 eV) at a pass energy of 23.5 eV and an X-ray power of 50 W. After deducting the Shirley-type noise, the spectra were matched with the CasaXPS software version 2.3.25 (Casa Software Ltd, Teignmouth, UK). The investigated region had a maximum lateral resolution of 0.7 mm. The literature [32,33,34,35,36,37] provides the binding energies (BEs) for all carbon and oxygen environments. In order to calibrate the BE scale, the CH_x_ component was adjusted to 285 eV.

### 2.4. Preparation of PRP

With slight adjustments, a platelet-rich plasma (PRP) was produced, as described in [38,39]. Blood was drawn from non-smoking, healthy women after receiving their informed consent. Platelet-rich plasma was collected and activated by a three freeze-thaw cycler after the blood was centrifuged in special tubes (Plasmolifting TM, Moscow, Russia). Growth factors produced from plasma were extracted by centrifugation at 12,000× *g* for 10 min at 4 °C, and then stored at 70 °C until further use. Each milliliter of PRP contained 102.5 mg of dry materials.

Before participating in the study, all subjects provided their informed consent for inclusion. The protocol was approved by the Ethics Committee of the Research Institute of Clinical and Experimental Lymphology–Branch of the Institute of Cytology and Genetics, Siberian Branch of the Russian Academy of Sciences (RICEL-branch of ICG SB RAS), in line with the Declaration of Helsinki (Identifier: N115 from 24 December 2015).

### 2.5. PRP Immobilization

Before PRP immobilization, all samples were sterilized for 45 min under UV light. It should be noted that the UV or VUV irradiation may induce the polymerization enhancing the immobilization [40]; however, our previous studies revealed that the irradiation of PCL-ref does not enhance the immobilization of PRP [41,42]. To induce the covalent attachment of PRP to the plasma-coated nanofibers, the samples were immersed for 15 min in a 2 mg/mL solution of 1-ethyl-3-(3-dimethylaminopropyl) carbodiimide (EDC) (Sigma Aldrich, 98%) in water. The samples were properly cleaned with PBS before being incubated with PRP for 15 min. The sample was carefully rinsed with PBS following the reaction. The sample designation was PCL-COOH-PRP.

### 2.6. Testing the Stability of PCL-COOH-PRP

The stability of immobilized PRP was evaluated by soaking the PCL-COOH-PRP samples (sample size 1 × 1 cm) in PBS solutions (200 µL) at 3 different pH values (4.8, 7.0, and 8.1) for 1 h, 2 h, 1 day, 4 days, and 8 days at 37 °C. The selection of these pH values was aligned with the typically observed pH of chronic and acute wounds. The resulting samples were labeled PCL-COOH-PRP-“XXh”-“pHXX” and PCL-COOH-PRP-“XXd”-“pHXX”, where “XXh” (or “XXd”) represents the number of hours or days the samples were immersed, and pHXX was the solution’s pH. The samples after immersion were analyzed by XPS.

## 3. Results

### 3.1. Plasma-Modified PCL Nanofibers

Figure 1 depicts the morphologies of PCL-ref and PCL-COOH. The deposition of the plasma polymer resulted in minor alterations to the nanofibers’ thickness but no major changes to their topography. On the contrary, the chemistry of the PCL-ref and PCL-COOH were extremely different. Water contact angle (WCA) experiments clearly demonstrated the effect of plasma coating. The PCL-ref displayed a WCA of 123°. The WCA was significantly reduced during the plasma layer deposition (down to 10.0 ± 0.5°). Polar group grafting was responsible for the enhanced wettability of plasma-coated PCL nanofibers. XPS analysis revealed quantifiable differences in the surface chemistries.

Table 1 displays the XPS-measured chemical compositions of PCL nanofibers (PCL-ref) and PCL-COOH samples. Using the high-resolution spectra of each component, the atomic percentages of the elements were determined. More details were obtained from the high-resolution spectra fitting presented in Figure 2. The XPS C1s spectrum of PCL-ref can be fitted with a sum of 3 components: hydrocarbons CH_x_ (BE = 285 eV), ether group C-O (BE = 286.4 eV), and ester group C(O)O (BE = 289.0 eV), as shown in Figure 2a. The FWHM of the C-O peak was set to 1.35 eV, while the FWHMs of the CH_x_ and C(O)O components were 1.1 and 0.85 eV, respectively. The fitting of the XPS C1s spectrum of PCL-COOH was completely different. PCL-COOH was fitted with the sum of 4 components: hydrocarbons CH_x_ (BE = 285.0 eV, used to calibrate the BE scale), carbon singly bonded to oxygen C-O (BE = 286.55 ± 0.05 eV), carbon doubly bonded to oxygen C=O/O-C-O (BE = 288.0 ± 0.05 eV), and C(O)O (BE = 289.1 ± 0.05 eV). Figure 2b displays the concentrations of all constituents.

Significant differences in the oxygen spectra of PCL-ref and PCL-COOH were also visible. Indeed, the PCL-ref exhibited two well-defined peaks with quite similar concentrations and a low FWHM (Figure 2c), while the PCL-COOH exhibited a broad spectrum with a higher FWHM for less well-defined peaks (Figure 2d).

### 3.2. Quantification of Immobilized PRP

The immobilization of PRP bonding led to significant compositional changes. PCL-COOH-PRP did, in fact, have a significant nitrogen concentration of 11.8 at %. The increase in the nitrogen concentration was accompanied by a significant decrease in the oxygen concentration. All nitrogen atoms have amide functions (N-C=O, BE = 400.2 eV, FWHM 1.7 eV). Additionally, 0.4% of sulfur was also detected. All these findings suggested that PRP was significantly incorporated into our layers. However, quantifying the percentage of the surface covered by PRP was not possible, neither from these simple observations nor from standard C1s curve fitting. As a result, we developed a novel method for quantifying PRP incorporated into the surface. Previously, this method was approbated for complex mixed polymers [35].

Using the CasaXPS software, the XPS C1s signal PCL-COOH-PRP was approximated by introducing a new line that replicated the form of the signal from the PCL-COOH. This method allowed for the quantification of the remaining plasma polymer surface. Finally, the C1s signal was fitted with a sum of 4 components: the plasma layer (PCL-COOH, BE = 285 eV), CH_x_ (BE = 285.0 eV, FWHM = 1.3 eV), C-N (BE=286.4 eV, FWHM = 1.5 eV), and N-C=O (BE = 288.2 eV, FWHM = 1.1 eV). The fitted PCL-COOH-PRP is shown in Figure 3a. Hence, more than 50% of the surface is covered by CH_x_, C-N/C-O, and N-C=O. The reliability of our fitting was confirmed by the fact that the atomic concentration of nitrogen was correlated with the N-C-O and C-N-C-O concentrations. Indeed, if one simply calculates the number of nitrogen atoms in the C-N groups, it will result in [C-N/C-O] × [C]/100% = 11.8 at %, i.e., perfectly matching the atomic concentration of nitrogen (Table 1). This methodology is what we have used to analyze all samples after immersion in different buffers. The variations of BE and FWHM for all introduced peaks (CH_x_, C-N/C-O and N-C=O) were less than 0.05 and 0.1 eV, respectively.

### 3.3. Stability of Immobilized PRP

The role of the immobilized growth factors and viable proteins from the solutions of PRP is very significant, and, as previously shown, the modification of polycaprolactone fibers with human platelet-rich plasma significantly increases the number of cells on the nanofibers, enhances their adhesion, and facilitates the wound healing process [25,27,42]. The presence of these viable molecules on the surface is highly important to maintain sufficient enhancement of the wound healing process facilitated by the nanofibers. Hence, when one is developing a wound healing material with superior properties, the stability of the immobilized biomolecules should be tested. It is also important to mention that the acidity of the wound might be affected by many parameters or circumstances, i.e., inflammation, diabetes, the acute or chronic course of the disease, etc. Depending on the parameters, the pH of the wound might vary from slightly acidic (pH 4–5) to slightly basic (pH 8). Thus, it is very important to understand the behavior of porosified materials with immobilized PRP in this pH range at a reasonable temperature (37 °C).

The atomic compositions of all samples are reported in Table 1. By following the concentration of nitrogen in the layer after soaking in the solutions, it is possible to have a semi-quantitative understanding of the remaining proteins and growth factors on the sample surface. For the sake of simplicity, the concentration of nitrogen is plotted as a function of immersion time in Figure 4a. The results indicated that the immobilized molecules remained on the surface for an extended period of time, as the nitrogen concentration was greater than 7% in all samples. However, the changes in the nitrogen concentration over the immersion time were significantly dependent on the pH. The decrease in the nitrogen concentration during the first hours for acidic and neutral pH was significantly lower than for the samples immersed at pH = 8.1. Interestingly, longer immersion of samples in the solution at pH = 8.1 led to significant recuperation of the nitrogen concentration, while samples immersed at a neutral pH exhibited a significant decrease in nitrogen. The variation of nitrogen concentration has a direct relationship with the amount of remaining proteins at the surface, but why it is pH dependent can be attributed to the properties (stability) of protein-surface linkage, the stability of plasma polymers, or the physical properties of immobilized proteins and growth factors. In order to better understand the behavior of our samples at different pH levels, the XPS C1s curve fitting was performed.

The XPS C1s curve fitting for all samples after immersion was performed using the same methodology presented in Section 3.2. The positions of CH_x_ and plasma polymer components had a fixed BE position of 285 eV. The positions of C-N/C-O and N-C=O were within the ranges of 286.35–286.45 and 288.1–288.2 eV, respectively. The spectra for the samples immersed in solutions at neutral pHs are depicted in Figure 3. The evolution of N-C=O (amide environment), plasma polymer, and CH_x_ (hydrocarbons from proteins or contaminations) contributions are shown in Figure 4b,c,d, respectively. The N-C-O contribution is a reasonably reliable indicator of the amount of immobilized biomolecules. Indeed, its position differs significantly from other peaks, and the amide contribution is directly attributed to peptide bonds in proteins, whereas the C-N/C-O contribution is related to alcohol groups and overlaps with the C-O contribution in the plasma polymer spectrum. The CH_x_ contribution can be both related to proteins and contaminants, as adventitious carbon is always present at the surface. Therefore, our attention was focused on the N-C=O evolution (Figure 4b). In contrast to the samples immersed at pH = 8.1, immersion at an acidic or neutral pH did not result in a significant decrease in N-C=O. Moreover, at a neutral pH, the samples exhibited some gain in the N-C=O percentage; however, the increase from 8.7 to 10.9% can be attributed to the heterogeneity of the samples or the low accuracy of such methodologies.

It is important to note that the decrease in N-C=O for the samples immersed at pH = 8.1 for 1 and 2 h recovered after a longer immersion time (1 day), similar to the evolution of nitrogen concentration. For very long immersion times, the remaining N-C=O percentage tends to behave regardless of the pH. After 8 days of immersion, all samples had very similar N-C=O percentages (and nitrogen concentrations). More importantly, even after 8 days, the concentrations of nitrogen and N-C=O were very high, significantly higher than the initial values (before the immersion). Hence, our materials exhibited very good long-term stability in a wide pH range.

## 4. Discussion

The reason for the significantly different behaviors during the first hours should be further discussed. The reason for different behaviors can be related to poor plasma polymer stability at high pHs. However, in this case, we should notice the features of PCL-ref and the lower plasma polymer contributions. In contrast, we have observed a very high increase in the percentage of plasma polymers for PCL-COOH-PRP-1h-pH8.1 and PCL-COOH-PRP-2h-pH8.1 (Figure 4c). Furthermore, the percentage of the plasma polymer contribution has been slightly increased at pH = 4.8. Hence, the decrease in N-C=O and nitrogen is not related to the faster dissolution of plasma polymers. A second possible reason could be the hydrolysis caused by the bonding of biomolecules with the surface. However, in this case, it is not possible to explain the recuperation of the nitrogen and N-C=O concentrations at a higher immersion time. Another explanation can be related to the physical properties of the proteins presented in the PRP.

PRP is a rich cocktail of various protein molecules, among which several growth factors are most important for regeneration and are measured in picograms. As known, the main component of PRP is albumin, which has an isoelectric point of pH = 4.9. At the isoelectric point, the protein is highly unstable in solutions and changes its conformation. At a higher pH, the protein becomes more stable in solutions, and its release into the solutions can be more probable. Therefore, our observations are most probably related to the hypothetical dynamics of the release of albumin since this protein is the most common protein in human blood plasma (more than 60% by mass). Albumin is a monomeric globular multicarrier of hydrophobic molecules such as fatty acids, hormones, growth factors, bilirubin, and fat-soluble vitamins. There is a dynamic structure to the interactions of albumin with other molecules (including growth factors), which are temporary, weak, multisite, and allosterically influence each other. At the isoelectric point, the protein is highly unstable in solutions and changes its conformation. At a higher pH, the protein becomes more stable in solutions, and its release into the solutions can be more probable. A possible reason for the difference in the dynamics of the content of amide bonds on the surface depends on the pH since its solubility significantly increases at an acidic pH. Hence, our immobilization procedure can be used to deliver therapeutic agents with a smart controlled release depending on the pH of the medium [43].

It should be noted that immobilization can occur not only towards the COOH groups of plasma polymers, but also to the pendent amino groups from proteins, as possibly in our case. Such bonding is less stable, but it can be released if it is stable in the solutions. Such a release would be more favorable at pH = 8.1 than at a lower pH. Hence, the decrease in nitrogen at pH = 8.1 in the first hours can be related to the faster release of “excess” PRP proteins. Later, these proteins can be redeposited onto the surface of the plasma polymer. At a lower pH, similar but much slower processes may occur.

As for the functional activity of proteins after covalent binding, we do not directly demonstrate the specific activity of immobilized proteins; however, indirectly, we confirmed that covalent immobilization significantly increases the proliferative activity of cells and their viability. In addition, we compared these results with ionic interaction, which also showed similar results but over a shorter period of time (within 3 days), with the subsequent leveling of the effect. In turn, covalently attached PRP on the 7th day demonstrates significant differences in contrast to unmodified nanofibers and fibers coated with PRP upon covalent binding [44].

The “reservoir” of PRP immobilized on a COOH plasma polymer layer may enhance the proliferation and migration of stem cells, attract macrophages, regulate the wound’s cytokine backdrop, and restrict inflammation. Promoting the formation of new capillaries accelerates epithelization in chronic wounds of various etiologies, hence enhancing wound healing.

## 5. Conclusions

The immobilization of PRP onto plasma polymers is a complex and challenging process. The quantification of grafted molecules can be performed using an advanced XPS fitting process, as shown in this paper. Our method revealed that more than 50% of the surface is covered by biomolecules. The release of biomolecules is dependent on the pH during the first hours of immersion. However, our nanofibers with immobilized PRP exhibited very sufficient long-term stability regardless of the pH, and, thus, they will have significant potential in future applications. Our approach, based on the use of autologous material (patients’ own blood plasma) and an FDA-approved polymer [45], and following successful in vivo experiments, showed that these materials have a high chance of obtaining approval for their use in clinical practice.

## Figures and Tables

**Figure 1 polymers-15-01440-f001:**
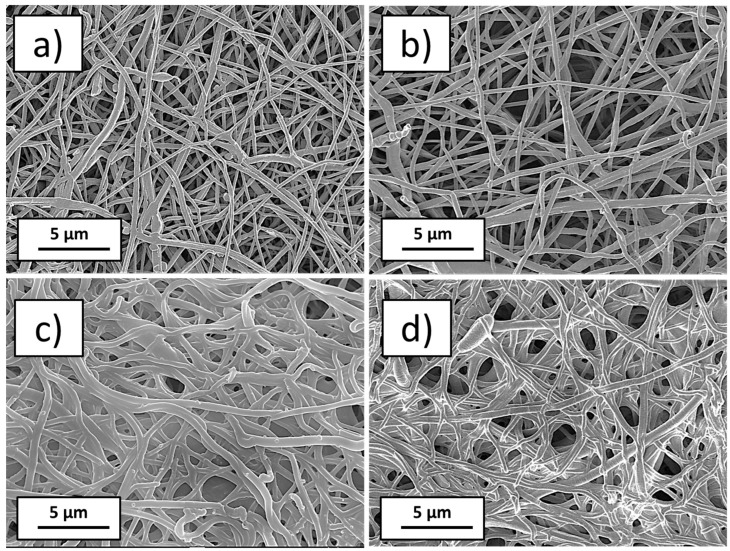
SEM micrographs at ×5000 magnification as prepared nanofibers PCL-ref (**a**), plasma-modified nanofibers PCL-COOH (**b**), plasma-modified nanofibers with immobilized PRP PCL-COOH-PRP (**c**), and plasma-modified nanofibers with immobilized PRP and after soaking in water at pH = 7, PCL-COOH-PRP-1d-pH7 (**d**).

**Figure 2 polymers-15-01440-f002:**
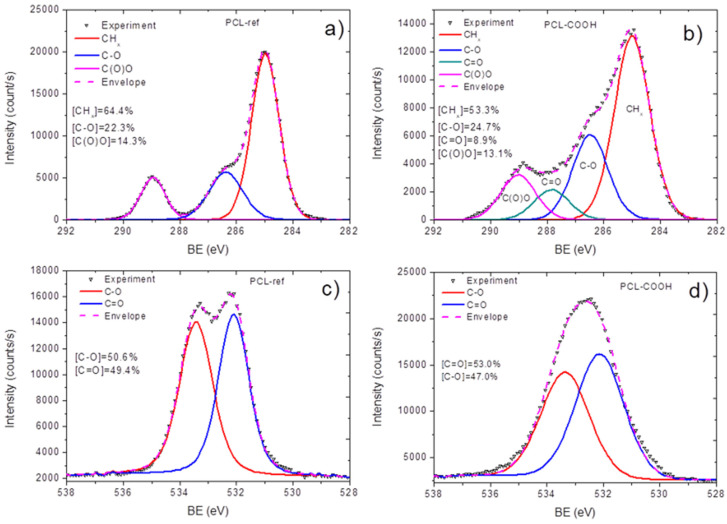
XPS C1s and O1s spectra of the samples: C1s curve fittings of as-prepared nanofibers PCL-ref (**a**) and plasma-modified nanofibers PCL-COOH (**b**); O1s curve fittings of as-prepared nanofibers PCL-ref (**c**) and plasma-modified nanofibers PCL-COOH (**d**).

**Figure 3 polymers-15-01440-f003:**
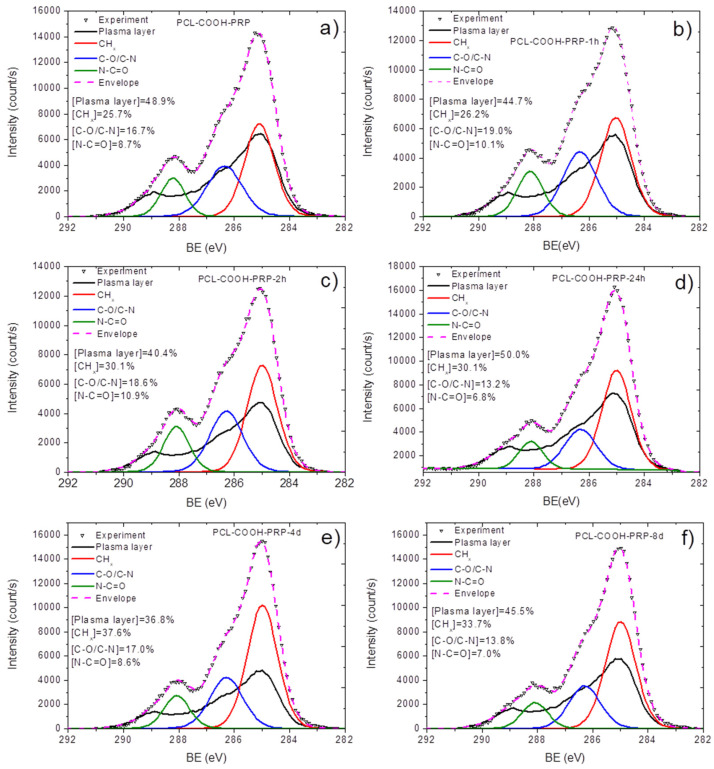
XPS C1s curve fitting of plasma-modified nanofibers with immobilized PRP as prepared PCL-COOH-PRP (**a**), after soaking in water at pH = 7 during 1 h PCL-COOH-PRP-1h-pH7 (**b**), after soaking in water at pH = 7 during 2 h PCL-COOH-PRP-2h-pH7 (**c**), after soaking in water at pH = 7 during 1 day PCL-COOH-PRP-1d-pH7 (**d**), after soaking in water at pH = 7 during 4 days PCL-COOH-PRP-4d-pH7 (**e**), after soaking in water at pH = 7 during 8 days PCL-COOH-PRP-8d-pH7 (**f**).

**Figure 4 polymers-15-01440-f004:**
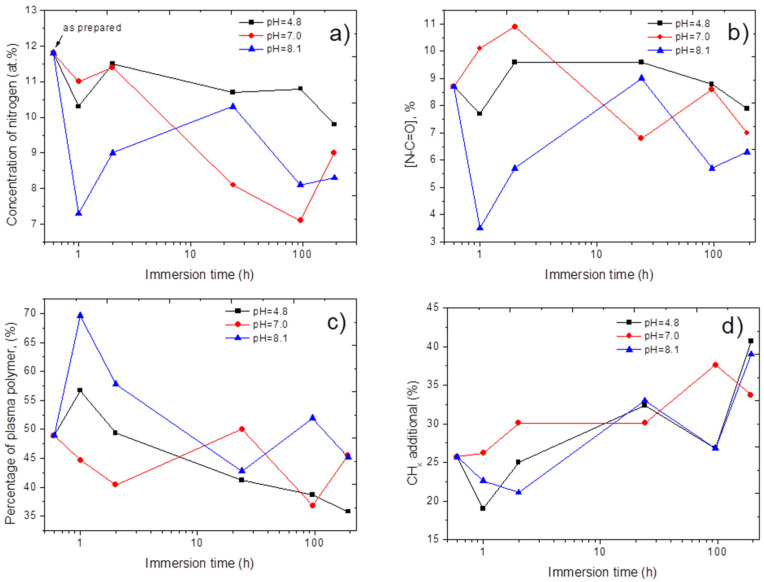
The evolution of nitrogen atomic concentration (**a**), concentration of amide contribution N-C=O (**b**), concentration of plasma polymer contribution (**c**), and hydrocarbons CH_x_ (**d**) environments derived from C1s XPS analysis as a function of immersion time at various pH.

**Table 1 polymers-15-01440-t001:** Atomic composition of samples determined by XPS analysis.

Sample	C	O	N	S
PCL-ref	77.3	22.7	0.0	0.0
PCL-COOH	69.3	30.2	0.5	0.0
PCL-COOH-PRP	70.7	17.2	11.8	0.3
PCL-COOH-PRP-1h-pH7.4	70.1	18.6	11.0	0.3
PCL-COOH-PRP-2h-pH7.4	70.6	17.6	11.4	0.4
PCL-COOH-PRP-1d-pH7.4	73.6	18.0	8.1	0.3
PCL-COOH-PRP-4d-pH7.4	74.7	17.0	7.1	0.3
PCL-COOH-PRP-8d-pH7.4	73.2	17.5	9.0	0.3
PCL-COOH-PRP-1h-pH4.8	70.0	19.4	10.2	0.4
PCL-COOH-PRP-2h-pH4.8	69.7	18.5	11.4	0.4
PCL-COOH-PRP-1d-pH4.8	72.1	17.0	10.6	0.3
PCL-COOH-PRP-4d-pH4.8	72.8	16.2	10.8	0.3
PCL-COOH-PRP-8d-pH4.8	74.1	15.9	9.8	0.3
PCL-COOH-PRP-1h-pH8.1	69.0	23.5	7.3	0.2
PCL-COOH-PRP-2h-pH8.1	69.1	22.0	8.7	0.2
PCL-COOH-PRP-1d-pH8.1	69.4	20.0	10.3	0.3
PCL-COOH-PRP-4d-pH8.1	70.0	21.7	8.1	0.2
PCL-COOH-PRP-8d-pH8.1	72.8	18.7	8.3	0.2

## Data Availability

Data are available from corresponding author upon a reasonable request.

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
