# Peer review of "Immobilization and Release of Platelet-Rich Plasma from Modified Nanofibers Studied by Advanced X-ray Photoelectron Spectroscopy Analyses"

_polymers, 2023, doi:10.3390/polym15061440_

Round 1

Reviewer 1 Report

The authors have presented a method for the surface oxidation of PCL nanofibers and have added PRP covalently to the the fibers. The surface coating was characterized with the use of XPS. The authors have proven the presence of PRP over 8 days and have discussed the degradation options.

Detailed opinion

Abstract

line 18: Please rephrase the sentence "regulated"

Introduction

line71: delete "very"

line 74: aqueous instead of "watery"

line92: Please rephrase the sentence "very away after treatment"

line123: Angiogenin

Materials Methods

line183: -70oC

line201: there are 2 dots at the end of the sentence

line216: analysis is the noun 

Discussion

The authors need to further explain that they cannot prove what type of proteins are present, most likely it is albumin, and they do not know if they are still functional after the covalent binding. Increased viability does not prove increased and/or specific functionality from ref [22,23].

Conclusion

The authors are too optimistic with the expression "they will have a high potential for both chronic and acute wound treatment" please keep in mind the regulatory related requirements.

Author Response

Answers to referee 1

First of all, we all are grateful to the reviewer for his positive feedback and great suggestions. Below are answers and corrections in point by point style

Comment 1

Abstract

line 18: Please rephrase the sentence "regulated"

It was substituted to revealed

Comment 2

Introduction

line71: delete "very"

Done

Comment 3

line 74: aqueous instead of "watery"

Done

Comment 4

line92: Please rephrase the sentence "very away after treatment"

Done

Comment 5

line123: Angiogenin

Corrected

Comment 6

Materials Methods

line183: -70oC

Space removed

Comment 7

line201: there are 2 dots at the end of the sentence

Done

Comment 8

line216: analysis is the noun 

Corrected

Done

Comment 9

Discussion

The authors need to further explain that they cannot prove what type of proteins are present, most likely it is albumin, and they do not know if they are still functional after the covalent binding. Increased viability does not prove increased and/or specific functionality from ref [22,23].

We have expanded and modified the Discussion part:

The reason for the significantly different behaviors during the first hours should be further discussed. The reason for different behaviors can be related to poor plasma polymer stability at high pH. However, in this case, we should notice the features of PCL-ref and the lower plasma polymer contributions. In contrast, we have observed a very high increase in the percentage of plasma polymer for PCL-COOH-PRP-1h-pH8.1 and PCL-COOH-PRP-2h-pH8.1 (Figure 4c). Furthermore, the percentage of plasma polymer contribution has been slightly increased at pH = 4.8. Hence, the decrease of N-C=O and nitrogen is not related to the faster dissolution of plasma polymers. A second possible reason could be hydrolysis caused by the bonding of biomolecules with the surface. However, in this case, it is not possible to explain the recuperation of the nitrogen and N-C=O concentrations at a higher immersion time. Another explanation can be related to the physical properties of the proteins presented in the PRP.

PRP is a rich cocktail of various protein molecules, among which several growth factors are most important for regeneration and are measured in picograms. As known, the main component of PRP is albumin, which has an isoelectric point of pH 4.9. At the isoelectric point, the protein is highly unstable in solutions and changes its conformation. At higher pH, the protein becomes more stable in solutions, and its release into the solutions can be more probable. Therefore, our observations are most probably related to the hypothetical dynamics of the release of albumin, since this protein is the most common protein in human blood plasma (more than 60% by mass). Albumin is a monomeric globular multicarrier of hydrophobic molecules such as fatty acids, hormones, growth factors, bilirubin, and fat-soluble vitamins. There is a dynamic structure to the interactions of albumin with other molecules (including growth factors), which are temporary, weak, multisite, and allosterically influence each other. At the isoelectric point, the protein is highly unstable in solutions and changes its conformation. At higher pH, the protein becomes more stable in solutions, and its release into the solutions can be more probable. A possible reason for the difference in the dynamics of the content of amide bonds on the surface depends on pH since its solubility significantly increases at an acidic pH. Hence, our immobilization procedure can be used to deliver therapeutic agents with a smart controlled release depending on the pH of the medium [43].

 It should be noted that the immobilization can occur not only towards the COOH groups of plasma polymers but also to the pendent amino groups from proteins, as possibly in our case. Such bonding is less stable, but it can be released if it is stable in the solutions. Such a release would be more favorable at pH = 8.1 than at a lower pH. Hence, the decrease of nitrogen at pH = 8.1 in the first hours can be related to the faster release of "excess" PRP proteins. Later, these proteins can be redeposited onto the surface of the plasma polymer. At lower pH, similar but much slower processes may occur. 

As for the functional activity of proteins after covalent binding, indeed, we do not directly demonstrate the specific activity of immobilized proteins; however, indirectly, we confirmed that covalent immobilization significantly increases the proliferative activity of cells and their viability. In addition, we compared these results with ionic interaction, which also showed similar results but over a shorter period of time (within 3 days), with the subsequent leveling of the effect. In turn, covalently attached PRP on the 7th day demonstrates significant differences in contrast to unmodified nanofibers and fibers coated with PRP upon covalent binding [44]

The "reservoir" of PRP immobilized on a COOH plasma polymer layer may enhance the proliferation and migration of stem cells, attract macrophages, regulate the wound's cytokine backdrop, and restrict inflammation. Promoting the formation of new capillaries accelerates epithelization in chronic wounds of various etiologies, hence enhancing wound healing.

Comment  10

Conclusion

The authors are too optimistic with the expression "they will have a high potential for both chronic and acute wound treatment" please keep in mind the regulatory related requirements.

We  have modified these sentences as follows:

PRP exhibited very sufficient long-term stability regardless of the pH, and, thus, they will have significant potential in future applications. Our approach, based on the use of autologous material (patients' own blood plasma) and an FDA-approved polymer [45] and following successful in vivo experiments, these materials have a high chance of obtaining approval for their use in clinical practice.

Reviewer 2 Report

The article “ Immobilization and release of PRP from modified nanofibers studied by advanced XPS analyses” by Anton M. Manakhov et al. is well-written, clear and coherent. However, the authors should improve the article along the following lines:

 Abstract

1)      Platelet-rich Plasma (PRP) is an ensemble of growth factors, extracellular matrix components and proteoglycans that are naturally balanced in the human body (or blood).

2)      “..from plasma-modified nanofibers..”  nanofibre surfaces modified by plasma treatment in a gas discharge.

Materials and methods

3)      “Before PRP immobilization, all samples were sterilized for 45 minutes under UV 191

Light”. It is well known that polymeric surface modification is achieved under UV irradiation (DOI:10.1063/1.3627160) and references therein. Please, add some references on  UV polymer surface modification and write a paragraph to convince the reader that surface modification is not via UV irradiation or  UV/ VUV irradiation produced by plasma, at least in this case.

Results

4)      Figure 2,  please, add additional information in the caption of Figures 1, 2, 3 and 4. The reader should be able to “read” the article only by looking at the figures.

References

The number of 35 references is limited to plasma and UV  polymer surface modification. Please,  add additional references.

Author Response

Answers to referee 2

First, we are all grateful to the reviewer for his positive feedback and excellent suggestions. Below are answers and corrections in point by point style

Comment 1

 Abstract

  • Platelet-rich Plasma (PRP) is an ensemble of growth factors, extracellular matrix components and proteoglycans that are naturally balanced in the human body (or blood).

Corrected accordingly

Comment 2

  • “..from plasma-modified nanofibers..”  nanofibre surfaces modified by plasma treatment in a gas discharge.

Corrected accordingly

Comment 3

Materials and methods

3)      “Before PRP immobilization, all samples were sterilized for 45 minutes under UV 191

Light”. It is well known that polymeric surface modification is achieved under UV irradiation (DOI:10.1063/1.3627160) and references therein. Please, add some references on  UV polymer surface modification and write a paragraph to convince the reader that surface modification is not via UV irradiation or  UV/ VUV irradiation produced by plasma, at least in this case.

We added a sentence:

It should be noted that the UV or VUV irradiation may induce the polymerization enhancing the immobilization[35], however, our previous studies revealed that the irradiation of PCL-ref does not enhancing the immobilization of PRP [36,37].

As regards of the comment related to or  UV/ VUV irradiation produced by plasma, we cannot neglect the effect of VUV-induced polymerization. For example, in https://doi.org/10.1016/j.radphyschem.2009.08.009 the deposition of thin films from gas mixtures by VUV was demonstrated. However, UV polymerization has a slower deposition rate as compared with our case. Also, the activation of CO2 molecules with the subsequent formation of COOH groups can be less efficient as compared to the reactions with plasma species.

Comment 4

Results

  • Figure 2,  please, add additional information in the caption of Figures 1, 2, 3 and 4. The reader should be able to “read” the article only by looking at the figures.

All Figure captions were modified/

Comment 5

References

The number of 35 references is limited to plasma and UV  polymer surface modification. Please,  add additional references.

We have added additional seven references.

Round 2

Reviewer 2 Report

The article can be published in its present form.